# Optimized procedure for the determination of alkylamines in airborne particulate matter of anthropized areas

Davide Spolaor[1], Lidia Soldà[1], Gianni Formenton[2], Marco Roverso[1], Denis Badocco[1],

Sara Bogialli[1], Fazel A. Monikh[1,3], Andrea Tapparo[1]

[1]Dipartimento di Scienze Chimiche, Università di Padova, via Marzolo 1, 35131 Padova, Italy

[2]ARPAV Environmental Regional Agency, Laboratory Department, via Lissa 6, 30171
  Mestre, Venice, Italy

[3]Institute for Nanomaterials, Advanced Technologies and Innovation, Technical University of
  Liberec Bendlova 1409/7, 460 01, Liberec, Czech Republic

**Correspondence**: Andrea Tapparo (andrea.tapparo@unipd.it)

**Abstract**

Due to their role in the formation of secondary aerosol, the concentrations of the most abundant aliphatic amines (methylamine (MA), dimethylamine (DMA), ethylamine (EA), diethylamine (DEA), propylamine (PA), and buthylamine (BA) present in the aerosol of a very anthropized area were measured by an optimized analytical procedure. PM10 samples were collected in the tanning district of Vicenza (in the Po Valley, northern Italy) in autumn 2020. Alkylamines were extracted in water and converted to carbamates through derivatization with Fmoc-OSu for subsequent determination by UHPLC with fluorescence detection. The procedure has been optimized obtaining very satisfactory analytical performances: LODs were in the range of 0.09-0.26 $ng/m^3$, average uncertainty of 3.4 % and recoveries of 95-101 %. The mean total concentration of the 6 amines measured in this study was $37 \pm 17$ $ng/m^3$, with DMA making the largest contribution. The proposed procedure may contribute to a better characterization of the local aerosol. In our preliminary investigation, the Pearson's correlation test showed that amines correlate strongly with each other and with secondary inorganic ions ($NH_4^+$, $NO_3^-$ and $SO_4^{2-}$), confirming that they compete with ammonia in the acid-base atmospheric processes that lead to the formation of nitrate and sulfate particles. The developed method allows to gather critical information about the load of aliphatic amines in PM to gain more insights into the sources and fate of these chemicals in the atmosphere.

# 1. Introduction


Atmospheric particulate matter (PM) is the subject of many discussions about the health of humans and ecosystems. A polluted atmosphere contains hundreds of organic and inorganic compounds involved in the formation of secondary aerosols and that contribute to their toxicity. Atmospheric amines could be especially important due to their acid neutralizing capacity. These

amines are derivatives of ammonia with one or more hydrogen atoms replaced by an alkyl or aryl group. Like ammonia, they are hydrophilic basic compounds that exist in the atmosphere in both gaseous and particulate phases. Amines could undergo rapid acid-base reactions with acids such as $H_2SO_4$ and $HNO_3$, similar to the transformation that occurs for ammonia itself (Ge et al., 2011b). Theoretical and experimental studies also show that alkylamines can react

with ammonium sulfate and ammonium bisulfate clusters resulting in the displacement of ammonium with aminium (Bzdek et al., 2010; Qui et al., 2011). Amines are also believed to play an important role in stabilizing sulfuric acid clusters in the early stages of new particle formation (NPF) (Loukonen et al., 2010; Almeida et al., 2013; Yao et al., 2018; Xiao et al., 2021). They can also contribute to the production of secondary organic aerosol (SOA) by

reacting with organic acids and oxidants (e.g. $O_3$ or $\cdot NO_3$) (Denkenberger et al., 2007; Tang et al., 2013; Nielsen et al., 2012; Malloy et al., 2009). Most amines are toxic and irritants to the eyes, skin, and mucous membrane and some of their oxidation products can also be carcinogenic (e.g. nitrosamines) (Lee and Wexler, 2013). The most common organic amines found in the atmosphere are C1-C6 alkylamines, which can be emitted from a variety of

anthropogenic and natural sources (Ge et al., 2011a). The main natural sources include biomass burning and biological activity, in particular from marine environments (van Pinxteren et al.; 2019, Xie et al., 2018). Animal husbandry is one of the main anthropogenic sources of atmospheric amines, along with various industrial activities, in particular the food industry. Other anthropogenic sources include fuel combustion (Shen et al., 2017), vehicular traffic,

composting, the treatment of industrial waste, and the combustion of tobacco (Ge at al., 2011a). Tanneries may also represent a significant source (JRC reference reports, 2013) as amines are formed and released in the atmosphere both in the preliminary steps of the tanning process (storage, desalting, and pealing of the row hide) and in the wastewater treatment plants. In urban areas anthropogenic nonagricultural sources could represent the main contribution to amines

emission (Shen et al., 2017; Cheng et al., 2020; Liu et al., 2022). Furthermore, emerging emission sources of amines have been associated with their use in the $CO_2$ capture processes

and, possibly, in catalytic processes for the reduction of NOx emissions from engines (Ge et al., 2011a; Muthiya et al., 2022; Stanciulescu et al., 2010)

Measurement of the ambient concentrations of atmospheric amines is challenging due to their high volatility and polarity. Common techniques for the determination of low molecular weight alkylamine are gas chromatography (GC), high-performance liquid chromatography (HPLC), and ion chromatography (IC), coupled with different detectors. In direct analysis by IC the separation and quantification of all analytes is often difficult due to interference from other cations (e.g., sodium or potassium) and the difficulty of separate some of the amines (e.g., diethylamine and trimethylamine) (VandenBoer et al., 2012; Huang et al., 2014; Zhou et al., 2018; Feng et al., 2020). These problems can be addressed by carefully adjusting the temperature and the elution program, or by modifying the retention times of the interfering ions with an eluent additive (Place et al., 2017; Feng et al., 2020). PTR-MS is the elective technique for the direct (and on-line) determination of amines in the gas phase (Wang et al., 2020; Chang et al., 2022) with elevated analytical performances (fast response, high sensitivity). Unfortunately, it fails the separation of amines with the same molecular mass (i.e. DMA/EA and PA/trimethylamine, TMA). This drawback is also reported in direct HPLC-MS analysis of amines (Kieloaho et al. 2013; Tzitzikalaki et al., 2021), in view of the difficulties in separating these species (non-derivatized, in the absence of a suitable MS$^2$ detection system). Direct analyses of amines can also be performed using GC (Bai et al., 2019). However, the polar nature of small aliphatic amines makes these analyses unfavorable due to the adsorption and decomposition of the solute in the column, resulting in peak tailing and analyte losses.

Because of the poor results often obtained with direct analyses of amines, the derivatization of the target analytes is recommended. Organic amines can be derivatized with various methods such as acylation, carbamate formation, Schiff base formation and silylation (Płotka-Wasylka et al., 2015). Among these methods, acylation and carbamate formation are the ones most commonly used because they allow a simple and fast derivatization step, where amines are easily derivatized in aqueous solutions and mild conditions. The derivatization is usually followed by a GC or HPLC separation. Derivatization can also be used to add a chromophore (or a fluorophore) to derivatized analytes, allowing both improved chromatographic performance and the detection by UV-vis absorption or fluorescence (Płotka-Wasylka et al., 2015; Szulejko and Kim, 2014) with significant higher sensitivities with respect to the conventional detection of non-derivatized species. In this work a simple and sensitive analytical method for the determination of aliphatic amines contained in atmospheric PM has been developed and validated. It makes use of a water extraction of amines, their derivatization by

Fmoc-OSu (9-fluorenylmethoxycarbonyl-N-hydroxysuccinimide), and their instrumental analysis by UHPLC with fluorescence detection (FLD). The procedure has been applied for the determination of six volatile aliphatic amines (methylamine (MA), dimethylamine (DMA), ethylamine (EA), diethylamine (DEA), propylamine (PA), and buthylamine (BA)) contained (as aminium salts) in PM10 samples of a highly anthropized area. The results are also preliminarily discussed to assess the contribution of aliphatic amines in the formation of secondary aerosol.

## 2. Materials and Methods

### 2.1 Chemicals

Ultrapure water (18.2 MΩ cm, ELGA LabWater PURELAB Chorus 1 Complete) and 37% HCl solution (ACS reagent, Riedel-de Haën) were used for the extraction of the analytes; Fmoc-OSu (9-fluorenylmethoxycarbonyl-N-hydroxysuccinimide, Sigma Aldrich), acetonitrile (>99,95%, Carlo Erba), sodium tetraborate (Sodium tetraborate decahydrate, Normapur) and NaOH (>97%, ACS reagent, Sigma Aldrich) were used for the derivatization step. Methylamine (MA), dimethylamine (DMA), ethylamine (EA), diethylamine (DEA), propylamine (PA) were purchased as hydrochloride salt (>98%, Sigma Aldrich) while buthylamine was purchased as pure liquid (99.5%, Sigma Aldrich).

### 2.2 Aerosol Sampling

The sampling campaign was carried out in the Chiampo Valley (Po Valley, northern Italy), an area located near Vicenza and characterized by the presence of numerous leather industries (tanning district). PM10 samples were collected for 24 h, and PM10 mass concentrations were directly measured using an automated beta attenuation analyzer (SWAM 5a Monitor, FAI Instruments, Italy) working at a constant flow rate of 2.3 m$^3$/h and equipped with 47 mm quartz filters (Whatman, QM-A quartz filters, Particle Retention Rating 98% at 2.2 μm). The filters were not treated before use, acceptable blank values were always obtained (OC<0.4 μg/m$^3$, DMA<<LOD). This sampling and measurement procedure (implemented in the air quality monitoring network of the Environmental Regional Agency, ARPAV) is the standardized "Ambient Air - Automated measuring system for the measurements of the concentration of particulate matter (PM10; PM2.5)" (EN 16450:2017). It is worth noticing that artifacts can occur in the sampling of aliphatic amines, as described by Shen et al. (2017). They are mainly

due to gas/particle phase transfers and some structure rearrangements, although different hypotheses regarding DEA are proposed. However, in studies in which gaseous amines are sampled on acidified quartz filters, significant absorption on untreated filters used for the PM sampling seems to be excluded (Chen et al.; 2022). We did not estimate these artifacts, but considering that amines are present in PM as aminium salts, the sampling losses of these low-volatile compounds were probably limited. In this regard, more in-depth analytical studies would be desirable.

The sampling campaign was conducted in three different areas of the Chiampo Valley over a two-month period, from the 18th of September to the 7th of October 2020 (site of Montebello), from the 13th of October to the 28th of October 2020 (site of Montorso) and from the 30th of October to the 18th of November 2020 (site of Trissino). A total of 53 PM10 samples were collected. Filter samples were kept at −20 °C until further analysis. Analysis of water-soluble ions (by ion chromatography), organic carbon, and elemental carbon (by Sunset OC/EC analyzer) was also performed using well-established procedures (Giorio et al., 2022; Khan et al., 2016).

## 2.3 Sample Preparation

Analytes were extracted from a portion of the filter (2/3) with 3 mL of a 0.1 M HCl solution in ultrapure water followed by sonication for 10 minutes at a temperature below 25 °C. 1.5 mL of the extract was then filtered (PTFE membrane, 0.2 μm pore size) inside a 1.5 mL test tube and stored for further analysis in a freezer at -18 °C. The derivatization reagent was prepared by dissolving a few mg of Fmoc-OSu in acetonitrile to obtain a concentration of 8.9 mM (3 g/L). Borate buffer (0.13 M) was prepared by dissolving 5 g of sodium tetraborate in 100 mL of milliQ water. An aliquot of 300 μL of the sample solution was added in a 1.5 mL test tube and neutralized by adding 5 μL of a 6 M NaOH solution. Then 150 μL of borate buffer and 250 μL of derivatization reagent were added to the solution (pH≈9). The mixture was stirred for about 30 seconds and allowed to react for 10 minutes at 50 °C using a thermostatic heating block.

## 2.4 Instrumental analysis

The solution containing the derivatized analytes was injected into an HPLC (LC-20AD XR Shimadzu; Canby, OR, USA) equipped with an autosampler (SIL-20AC XR Shimadzu, injection volume 5 μL) and a fluorescence detector (RF-20° XS Shimadzu). Excitation and emission wavelengths were 264 and 313 nm, respectively. The separation was performed using a Luna Omega 1.6 μm Polar C18 column (100 x 2.1 mm), with a constant flow rate of 0.2

mL/min and a column temperature of 30°C. A gradient elution program using acetonitrile (eluent B) and water (eluent A), to which formic acid was added (0.05%) to both solutions, was optimized and used to separate the derivatized amines. The elution program started with a mixture containing eluent B at 40%, after 1.5 min the concentration of B was increased linearly to 60% in the span of 13.5 min and kept constant for 4 min, then the concentration of B was raised to 100% in 0.2 min and maintained for 6 min (before returning the system to the initial condition, for a chromatographic run lasting 32 min). Concentrated standard solutions (1 g/L, for DMA it corresponds to 22 mM) were prepared by dissolving the hydrochloride salt of the amines in ultrapure water or, in the case of butylamine, by diluting the pure amine. From these solutions, a mixed working standard solution containing all amines was then prepared, each at 10 mg/L. This solution was diluted to obtain calibration solutions at suitable concentrations (10, 50, 100, 250 μg/L). These standards were analyzed in duplicate to obtain the calibration functions. Laboratory blanks consisting of ultrapure water and matrix blanks consisting of aqueous extract of blank quartz filters were treated and analyzed each day as regular samples to control the contribution of both chemicals and filters to the analytical signals. The real sample results were corrected for the matrix blank concentrations.

## 2.5 Analysis *via* ESI-QTOF-MS

The possible presence of chromatographic interferences was investigated by analyzing PM10 samples using independent detection techniques. Specifically, three selected PM10 samples were analyzed using the same sample preparation procedure and the same chromatographic method described in 2.3 and 2.4, but with two different detection systems: fluorescence and ESI-QTOF-MS (Agilent 6545, mass resolution 35000). The UHPLC-MS analysis was performed with an ESI source, operating in positive acquisition mode. Source parameters were: 300 °C capillary temperature, 40 a.u. sheath gas flow, 300 °C sheath gas temperature, 20 a.u. auxiliary gas flow, 350 °C auxiliary gas temperature, 4 kV spray voltage. Data were acquired in the range of 100-1000 *m/z* (scan rate).

## 2.6 Statistical analysis

The correlations among the data acquired during the campaign (temperature, relative humidity, atmospheric precipitation, concentration of atmospheric pollutants - $NO_x$, $H_2S$ and $NH_3$ - Table S1) and the measured concentrations of PM10 components (elemental and organic carbon, inorganic ions and amines; Tables S2, S3 and S4) were evaluated by the Pearson's correlation coefficients. Principal component analysis (PCA) was also performed considering the full set

of PM10 samples and species analyzed. Atmospheric parameters such as temperature, relative humidity, and precipitation quantities were used as supplementary variables.

## 3. Results and Discussion

### 3.1 Optimization of the analytical procedure

The aliphatic amines in the atmospheric particulate matter were determined here by extracting the analytes from the sampling filters and subsequently analyzing by UHPLC-FLD, after a derivatization procedure using Fmoc-OSu. Details regarding these reactions are reported in the literature (Iqbal et al., 2014; Lozanov et al., 2004; Płotka-Wasylka et al., 2015). The advantages of this method consist primarily of its rapidity and simplicity; the amines are readily derivatized

in an aqueous alkaline solution with the addition of Fmoc-OSu and are directly injected into the HPLC column without the need for further extraction steps.

It is worth noting that the derivatization step is generally mandatory for the separation of amines via HPLC or GC (Huang et al., 2014) and most derivatizing agents are suitable only for primary and secondary amines, thus precluding the analysis of tertiary amines. However, the

220 derivatization of TMA is possible using Fmoc, but the reaction is very slow (Szulejko et al., 2014), especially in aqueous solutions. As a consequence, only primary and secondary amines can be determined by the present procedure. This is confirmed by the fact that, after adding TMA to a standard solution or to a sample, no peak attributable to TMA was identified in our chromatograms. Moreover, regarding a possible bias in the quantification of DMA (tertiary

amines derivatized with Fmoc may dealkylate, yielding a product identical to the derivatized secondary amine), no difference in the peak intensity of DMA was observed.

**3.1.1 Optimization of chromatographic conditions.** Three RP-C18 columns were preliminary tested for the UHPLC separation of the six derivatized aliphatic amines (Shimadzu

XR-ODS III C18, Phenomenex Luna Omega polar C18, Phenomenex Kinetex C18) showing similar performances in terms of selectivity. The Luna Omega column was chosen because of its slightly higher efficiency (better peaks shape). The elution program (gradient elution using acidic water/acetonitrile, for a chromatographic run lasting 32 min) has been optimized to allow the proper separation of each amine, in particular the ethylamine/dimethylamine and

butylamine/diethylamine pairs. The correct separation of methylamine was also problematic due to the presence of a large peak attributable to the by-products of Fmoc-OSu and to the

derivatized $NH_3$, which are both present in the solution in large excess. The addition of formic acid in the two solvents (0.05%) did not have an effect on the retention time of the amines, but slightly reduced the retention time of the Fmoc by-products. Changes in the chromatographic method were aimed at reducing the elution time without reducing the separation efficiency. Figure 1 shows representative chromatograms obtained for a standard solution (100 µg/L) and a PM10 sample extract (blank sample chromatogram is reported in Figure S1). The order of retention times of Fmoc derivatives is as follows: methylamine (MA, $t_r$=10.1), ethylamine (EA, $t_r$=12.4), dimethylamine (DMA, $t_r$=13.6), propylamine (PA, $t_r$=14.8), buthylamine (BA, $t_r$=17.4), and diethylamine (DEA, $t_r$=18.1); the order is the same as that reported in other articles (Shen et al., 2017; Cheng et al., 2020).

**3.1.2 Optimization of extraction and derivatization steps**. Compared to the procedures of Shen et al. (2017) and Cheng et al. (2020) we added a larger amount of derivatizing agent (250 µL in 700 µL (3.2 mM) instead of just 400 µL in 10 mL (0.36 mM) of Fmoc-OSu in acetonitrile) obtaining stable peaks for each of the six target amines. Therefore, the extraction and reaction steps (time, temperature, and reaction mixture composition) were optimized to gain the maximum recovery for each of the analytes. The extraction was initially tested with ultrapure water, but the addition of hydrochloric acid (0.1 M) showed better results in terms of recoveries (from 80-95% to >95%), due to the higher water solubility of the aminium ions with respect to the non-protonated amines.

The extraction time was initially set at 20 min, but no significant differences in instrumental signals were observed after reducing it to 10 min (Table S5). The derivatization step was tested at different temperatures (25, 50, 60 °C) and reaction times (5-20 min) (see Table S6 as an example). Using a large excess of Fmoc-Osu, the reaction was quantitative (also for DEA) and very fast even at ambient temperature, as it takes less than 5 minutes to complete.

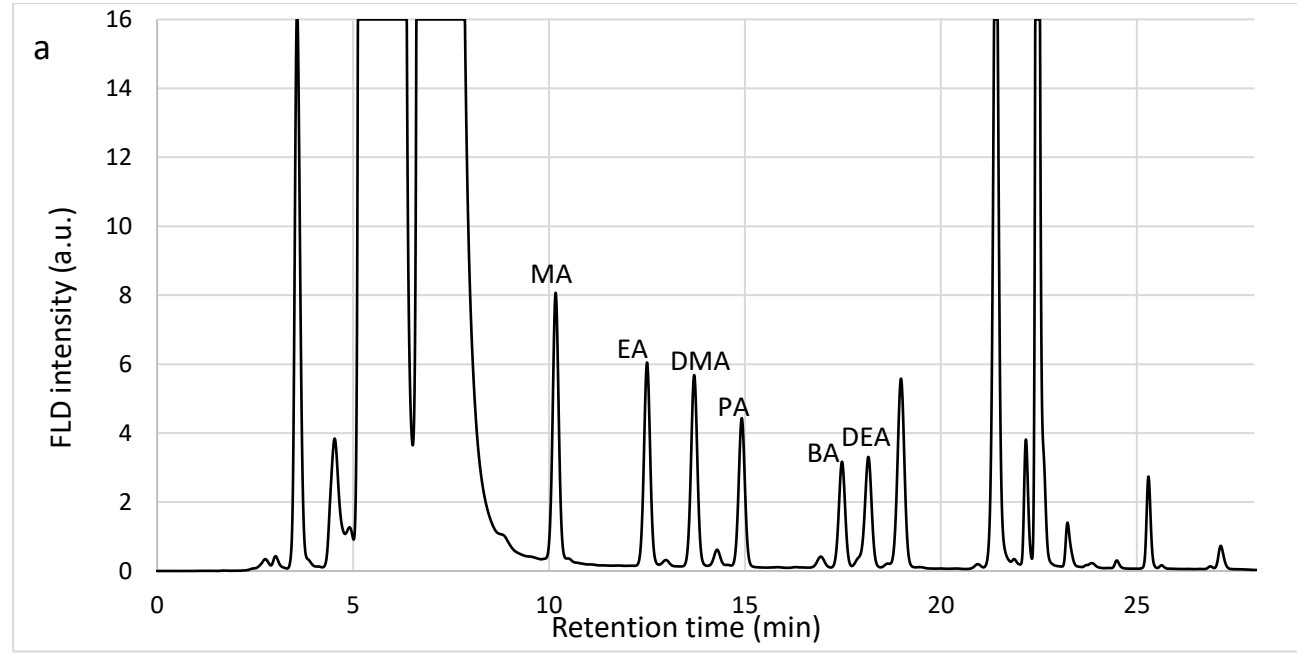

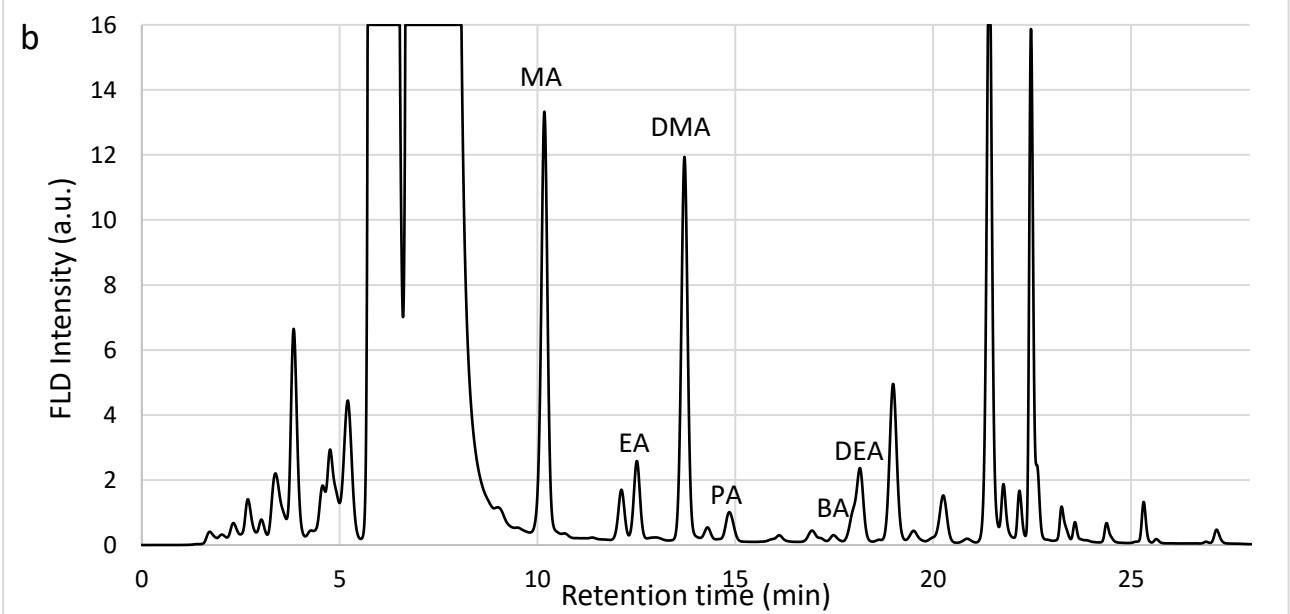

**Figure 1.** *UHPLC-FLD chromatograms of (a) standard solution of the six amines at 100 µg/L each and (b) a PM10 sample extract (sample of 02.10.2021, PM10=24.5 µg/m³*

However, because the reaction time and temperature affected the amount of unreacted Fmoc in the solution (that may represent a serious chromatographic interference), the reaction was carried out at 50 °C, for 10 min, to completely eliminate the chromatographic peak relative to the unreacted Fmoc that coelutes with the derivatized ethylamine (Table S6). The same results were also obtained using Fmoc-Cl as the derivatizing agent. In conclusion, if compared with most previous published procedures that use a derivatization reagent (Shen et al., 2017; Cheng et al., 2020; Choi et al., 2020; Akyüz, 2008; Majedi and Lee, 2017; Parshintsev et al., 2015),

our sample pre-treatment does not need successive liquid-liquid extraction or preconcentration steps.

**3.2 Method validation**

The linearity of the calibration functions, limits of detection (LOD), precision (repeatability), and recoveries of the method were measured experimentally. The calibration functions of the six aliphatic amines (ranged from 0 µg/L, the solvent blank, to 250 µg/L) were all assessed as linear after an F test between the linear and quadratic regression variance ($\alpha = 0.05$; $F_{(0.05,8,7)} = 3.73$; $F_{sp} < 1.85$; $p > 0.2$). Repeatability was evaluated as the relative standard deviation

(RSD,%) over four equal parts of the same sample (a PM10 filter), each spiked with 15 µL of a mixture containing the six amines at 10 mg/L (corresponding to 4.1 ng/m$^3$ in the sampled PM10 and in line with the concentration ranges measured in the real samples). The measured RSD were less than 5% for all the amines. The recoveries were evaluated by adding different quantities of a mixture containing each of six amines at 10 mg/L (0, 20, 40, 60 µL) to four equal

parts of the same PM10 sample and determining the slope of each recovery function ($C_{experimental}$ *vs* $C_{added}$). It was found that the recoveries ranged from 95% to 101%, supporting the hypothesis of a negligible effect (bias) of the matrix components on the effective recovery of each analyte at various steps of the procedure, as well as on the instrument response. The limits of detection (LOD) were estimated from the calibration function using the Hubaux-Vos method (Hubaux

and Vos, 1970), obtaining values ranging from 1.1 to 3.2 µg/L, which correspond to an atmospheric concentration of 0.088-0.26 ng/m$^3$. These validation parameters are listed in Table 1.

These performances, compared to the ones obtained by previously developed methods (Huang et al., 2014; Choi et al., 2020; Parshintsev et al., 2015), are characterized by similar or higher

LODs, mainly due to lower sampling volume and the absence of a preconcentration step, but also by quantitative recovery functions, and good repeatability. Furthermore, our optimized sample preparation procedure allows for the correct quantification of DEA that was not carried out using analogous methods (Shen et al., 2017; Cheng et al., 2020).

*Table 1 Detection limit, recovery, and relative standard deviation of the analytical method developed in this study*

| Analyte | Detection limit (µg/L) | Detection limit (ng/m$^3$) | RSD (%)* | Recovery (%) |
|---|---|---|---|---|
| Methylamine | 2.79 | 0.23 | 3.89 | $100.7 \pm 6.0$ |
| Ethylamine | 1.72 | 0.14 | 3.54 | $99.1 \pm 3.9$ |
| Dimethylamine | 2.45 | 0.20 | 4.72 | $98 \pm 16$ |
| Propylamine | 1.24 | 0.10 | 3.58 | $100.4 \pm 2.3$ |
| Buthylamine | 3.16 | 0.26 | 3.19 | $95.2 \pm 2.0$ |
| Diethylamine | 1.07 | 0.09 | 1.56 | $97.0 \pm 6.5$ |

*estimated by the analysis of a real PM10 sample (spiked, added concentration 4.1 ng/m$^3$ of each amine, n=4)*

**3.2.1 Comparison of the results obtained by independent methods.** Three samples were analyzed with two different methods as described in section 2.5. In the MS analysis, the signal of the parent ion was used to identify each of the derivatized amines, but it was too low to be used for quantification. As a consequence, the MS signal at *m/z* 179.0861 (common to each amine, due to the in-source fragmentation of the derivatized analytes and attributable to a residual moiety of the derivatizing agent) was selected for the quantitative MS analysis. Butylamine was used as the internal standard, as it was not present in the analyzed samples. To ensure that each amine was present at a concentration high enough to be quantified, 10 µL of a mixture containing the amines at 10 mg/L was added to the three samples (corresponding to 1.8 ng/m$^3$). From the comparison of the results obtained by applying the two different methods (Table S7), it was found that the measured concentrations of the amine derivatives were statistically equivalent and that there was no significant systematic difference between the two detection methods. This result suggests the absence of significant chromatographic interferences (bias) in the optimized UHPLC-FLD analytical method.

**3.3 Environmental concentrations of aliphatic amines**

Table 2 shows the mean concentrations of the amines contained in PM10 samples collected in the three sampling sites considered in this study (Montebello, Montorso, Trissino, Figures S2-S3). Taking into account the average concentration (± standard deviation) throughout the sampling campaign (53 total samples), the amines present at higher concentrations were dimethylamine (19.0 ± 8.7 ng/m$^3$), methylamine (9.7 ± 5.5 ng/m$^3$) and diethylamine (4.5 ± 2.0 ng/m$^3$) while butylamine signal was below the limit of detection and therefore was not

considered in the present discussion. Absolute concentrations in the three sites of EA, PA, and DEA varied very little, while concentrations of MA and DMA were characterized by much wider distributions. The concentration of most of the amines did not show great variations between the three locations considered, except for dimethylamine, where the average concentration ranged from 13.7 ng/m$^3$ in Montebello to 24.9 ng/m$^3$ in Trissino (details in the box plots, Figure S4). The concentrations of aliphatic amines measured in this work are consistent with those described in other studies (Choi et al., 2020; Akyüz, 2008; Feng et al., 2020; Cheng et al., 2020), being in the range between a few tenths of ng/m$^3$ and 1 ng/m3. These studies are here briefly discussed as examples regarding the alkylamine concentrations commonly found in the PM of urbanized areas; more information is summarized in Table S8. In the work of Choi et al. (2020), the concentration of amines (EA, DMA, DEA, PA, BA) was measured in PM2.5 in the metropolitan area of Seoul (South Korea). The sum of the annual mean concentrations of the target amines was $5.56 \pm 2.76$ ng/m$^3$ with a low dependence on the sampling season. Consistent with our study, the main contribution was attributed to DMA, while the lowest was attributed to EA. However, the concentrations measured in Seoul were much lower than those measured in the Chiampo Valley (Vicenza-IT, present study). Cheng et al. (2020) measured EA, DMA, and DEA in Yangzhou, China. They obtained much higher values for EA, $12.58 \pm 10.45$ ng/m$^3$, but lower for the other two amines. Akyüz (2008) measured amines in PM10 sampled in the Zonguldak industrial area in Turkey, where pollution is very high due to the production and consumption of coal. Concentrations in the Zonguldak industrial area were rather high for all amines considered (7-9 ng/m$^3$) and higher during winter. In the work of Feng et al. (2020), which was carried out in Shanghai, the amines present at the highest concentrations were found to be DMA and MA, which reached average concentrations around 20 ng/m$^3$. The differences between the measured concentrations in this study and those reported in literature are probably due to the different emission sources present in the Chiampo Valley compared to the other locations. In particular, DMA appears to be present in large quantities in the Chiampo Valley compared to other amines, while BA was almost not detected (<1 ng/m$^3$). The high total concentration of amines could be attributed to the presence of significant emission sources, such as vehicular exhaust or intense industrial activity (tanning industry, in particular). Another potentially important source could be agricultural activity, which is also strongly associated with ammonia emission (Backes et al., 2016). However, recent studies suggest that in urbanized areas the contribution of non-agricultural sources is dominant (Chang et al., 2022). In this connection, we underline that since the 1960s the tanneries of the district have been the main responsible for the contamination of both the surface/ground waters and the

atmosphere of a larger area. In particular, high amounts of $H_2S$ and VOCs (mainly toluene, xylenes, ethylacetate, and butylacetate) are emitted into the atmosphere by this industrial activity. This is confirmed by the Regional Environmental Agency (ARPAV) which recently measured high atmospheric levels of $H_2S$ (up to 241 µg/m$^3$, daily average, and 1067 µg/m$^3$, hourly average) and estimated that 60-88% of the total VOC emitted comes from the tanneries of the district (ARPAV, 2023; JRC, 2013). Primary aerosol emissions from these tanneries are comparable to those estimated for other common sources, but, associated with high $H_2S$ and VOC emissions, a significant contribution to secondary aerosol production in a larger area is predictable. In effect, the significant levels of amines measured here would also be associated with the accumulation of pollutants that affects the entire Po Valley during Winter and Autumn (Masiol et al., 2015).

*Table 2* *Amines average concentrations and standard deviations in the three sites and during the entire campaign (53 total samples)*

|  | Montebello (n=19) | | Montorso (n=15) | | Trissino (n=19) | | Total (N=53) | |
|---|---|---|---|---|---|---|---|---|
|  | mean (ng/m$^3$) | SD (ng/m$^3$) | mean (ng/m$^3$) | SD (ng/m$^3$) | mean (ng/m$^3$) | SD (ng/m$^3$) | mean (ng/m$^3$) | SD (ng/m$^3$) |
| MA | 18.5 | 5.2 | 10.9 | 5.9 | 11.0 | 5.0 | 9.7 | 5.5 |
| EA | 2.6 | 1.2 | 2.5 | 1.4 | 1.95 | 0.76 | 2.4 | 1.2 |
| DMA | 14.2 | 5.6 | 19.4 | 9.3 | 24.9 | 7.2 | 19.0 | 8.7 |
| PA | 1.5 | 1.1 | 1.8 | 1.5 | 2.1 | 1.2 | 1.8 | 1.2 |
| DEA | 4.3 | 2.1 | 5.5 | 2.2 | 4.0 | 1.5 | 4.5 | 2.0 |

**3.4 Correlation among amine concentrations and other atmospheric variables**

Regarding the contribution that this method can offer in the study of atmospheric phenomena, in our preliminary study we observed that the amines contained in PM10 samples correlate rather strongly between themselves (Pearson's correlation coefficients are reported Table S9), indicating that their presence in PM is due to similar processes, i.e. the rapid acid-base reactions with atmospheric acids as described in previous works (Ge et al., 2011a). The concentration of all amines correlates strongly or moderately with PM10 (r>0.55, except ethylamine r=0.38), as expected for the aerosol components associated with the secondary fraction (that is significant in Po Valley). In effect, amines are also strongly correlated with nitrate (r>0.52) and ammonium (r>0.51) supporting the hypothesis that amines are present in the particulate mainly as a result of acid-base processes that lead to the formation of sulfates and nitrates.

Regarding the sources of emission, it is worth noting that DMA (the amine present at higher concentrations) shows a strong correlation with potassium (r = 0.73) supporting the hypothesis that biomass combustion processes are significant sources of amines. This hypothesis is further confirmed by the strong correlation (r = 0.80) between DMA and OC (organic carbon concentration in PM10), taking into account that both OC and potassium in PM are closely associated with biomass burning emissions (Scotto et al., 2021; Masiol et al., 2020). However, OC could also be related to industrial emission sources, which then would also contribute to the emission of DMA. These strong correlations have not been found for some amines. For example, significantly lower correlations of EA with both DMA and PM10 were estimated, possibly due to a specific source of EA (for example, traffic combustions) that differs from those that characterize other amines (biomass combustion, tanneries). Of course, this aspect is of great interest and needs further investigation. In this connection, the acquisition of more complete datasets (Masiol et al., 2020) that could include both traffic biomass combustion markers and the use of adequate source apportionment methodologies (PMF) will provide more detailed results regarding the amine sources of this area.

The PCA loading plot depicted in Figure 2 shows that PC1, which describes 53.22% of the total variance, is linked to the fine component of the atmospheric particulate, especially to the component of secondary origin, which includes all the amines. PC2, which expresses 14.98% of the variance, appears to be more related to the coarse component of the particulate matter. The loading plot confirms the strong correlation between organic carbon, PM10, nitrate, and sulfate, as well as with different amines, in particular with DMA and MA. The graph also shows that the gaseous substances ($NO_x$, $NH_3$, i.e. the precursors of the secondary compounds) do not correlate at all with the secondary components of the particulate. This result indicates that the concentration of these pollutants does not significantly affect the local concentration or the composition of PM10 on a daily basis, which is influenced mainly by meteorological parameters and primary emission sources (Masiol et al., 2017).

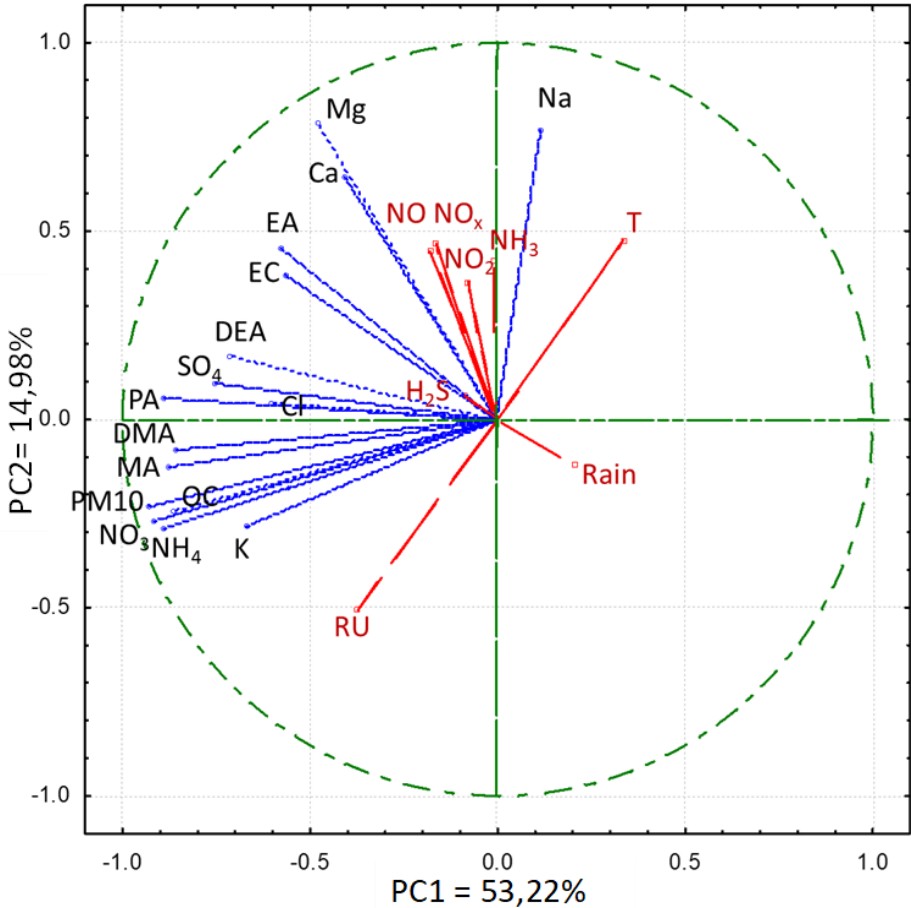

*Figure 2. PCA loading plot referred to the first two principal components. Blu vectors indicate active variables and red vectors refer to supplementary variables (score plot in Figure S5).*

## 4. Conclusions

The optimization of a fast sample preparation procedure followed by a UHPLC-FLD analysis allowed us to accurately determine the main alkylamines in the aerosol samples. The proposed analytical method performs well in terms of precision, detection limit, recoveries, and linear response range. The concentrations of amines measured in PM10 sampled in the tanning district

(Chiampo Valley, Italy) are comparable to those quantified in other urbanized sites, possibly associated with the presence of important local emission sources, such as vehicular traffic, intense industrial activity, and biomass burning. These sources may have contributed, in particular, to the production of dimethylamine. Amine concentrations are strongly correlated with PM10 and with secondary inorganic ions, supporting the hypothesis that amines are

present in the particulate mainly as a result of acid-base processes that lead to the formation of nitrates and sulfates. Dimethylamine, which is the amine present at higher concentrations, shows strong correlations with both the organic component of PM10 and potassium, indicating

the relevance of biomass combustion process emissions. We expect that the procedure developed here will find application in extensive environmental studies. In this connection, a promising perspective for the UHPLC-FLD method is the possibility to validate the analytical data obtained by the online aerosol analyzer (ATOF-MS, AMS) (Malloy et al., 2009; Bottenus et al., 2018; Huang et al., 2012; Giorio et al., 2015).

**Author contributions**

DS and AT are the principal investigators; they are responsible for all research activities. LS supported the laboratory analysis of PM samples. GF coordinated the sampling campaigns and performed the OC/EC analysis. MR and SB were responsible for LC-MS analysis. DB was involved in the statistical analysis of the results. FAM contributed to the interpretation of the data and to the text revision.

**Acknowledgements**

We acknowledge the support from ARPA Veneto, Dept. of Vicenza (IT), in the PM10 sampling campaign.

**Competing interests**

The authors declare that they have no conflict of interest.

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
