# Peer review of "Optimized procedure for the determination of alkylamines in airborne particulate matter of anthropized areas"

_Aerosol Research, 2023_

## Author Response (AR1)

**AR-2023-3 - Anonymous Referee #1 Comments and *Reply by the Authors* (*italic*, actions in blue)**

The authors developed a procedure to identify and quantify primary and secondary alkylamines in airborne particulate matter. They applied the developed method to quantify 5 alkylamines in PM10 samples collected in North Italy, and they try to investigate the sources of emission through a statistical approach. The data about the emission are not conclusive, however I think the main scope of this work is to develop and validate an analytical method rather then a study on sources and concentrations of amines in the atmosphere. The manuscript is clear and well-written, the procedure developed is simple and effective and in future it could be easily applied to a larger number of samples.

Here are some minor comments:

Line 113-114: were the filters treated before use? Please specify it.

*The filters were not treated before use. In fact, very low blank values were always obtained, for instance OC<0.4 µg/m3, DMA<<LOD.  A proper phase has been added:*

… equipped with 47 mm quartz filters (Whatman, QM-A quartz filters, Particle Retention Rating 98% at 2.2 µm). The filters were not treated before use, acceptable blank values were always obtained (OC<0.4 µg/m$^3$, DMA<<LOD).

Lines 119-125: mid-September is not always the cold season and the temperature difference between September and November could be relatively high. Indeed, the temperature during the campaign period ranged between 24 and 8 degree C. Please comment this in regard to the artefacts and sampling losses of volatile and semivolatile compounds.

*The authors thank the reviewer for evidencing this important aspect. Although the temperature changed in the range 8-24 °C, we think that artefacts in amines determination were very limited. In fact, we focused on amines, but in PM we find their ammonium salts, rapidly formed in the atmosphere by reaction with acids, which characterized by a significantly lower volatility. The relevant phase has been rewritten.*

It is worth noticing that artifacts can occur in the sampling of aliphatic amines, well described by Shen et al. (2017). They are mainly due to gas/particle phase transfers and some structure rearrangements, although different hypotheses regarding DEA are proposed. We did not estimate these artifacts, but considering that amines are present in PM as ammonium salts, the sampling losses of these low-volatile compounds were probably limited.

Lines 139-140: was the test tube placed in a bath or stored in an oven? Please give more details about that.

*The requested details have been provided.*

… and allowed to react for 10 minutes at 50 °C using a thermostatic heating block.

Paragraph 2.4: Could you please provide the retention times of the derivatized amines? Could you please provide the resolution of the MS instrument?

*Now, in section 3.1.1 are detailed the retention times of the analytes; in section 2.5 we added the Mass resolution value of the MS detector.*

The order of retention times of Fmoc derivatives is as follows: methylamine (MA, $t_r$=10.1), ethylamine (EA, $t_r$=12.4), dimethylamine (DMA, $t_r$=13.6), propylamine (PA, $t_r$=14.8), buthylamine (BA, $t_r$=17.4), and diethylamine (DEA, $t_r$=18.1);

… ESI-QTOF-MS (Agilent 6545, mass resolution 35000). The UHPLC-MS analysis was performed with …

Paragraph 3.1 - Lines 192-199: It's a bit confusing this part. Also if the derivatization of tertiary amines is possible using Fmoc, the procedure you developed is not suitable for tertiary amines. Please clearly states that the analytical procedure you developed is suitable only for primary and secondary amines.

*The relevant phrase has been rewritten.*

However, the derivatization of TMA is possible using Fmoc, but the reaction is very slow (Szulejko et al., 2014), especially in aqueous solutions. As a consequence, only primary and secondary amines can be determined by the present procedure. This is confirmed by the fact that, after adding TMA to a standard solution or to a sample, no peak attributable to TMA was identified in our chromatograms. Moreover, regarding a possible bias in the quantification of DMA (tertiary amines derivatized with Fmoc may dealkylate, yielding a product identical to the derivatized secondary amine), no difference in the peak intensity of DMA was observed.

Lines 211-213: Both eluents have been acidified with formic acid? Please specify it also in the paragraph 2.4. How it is written now seems that formic acid was added only to water.
*Many thanks, this lack has been corrected in Section 2.4.*
… using acetonitrile (eluent B) and water (eluent A), to which formic acid was added (0.05%) to both solutions,

Paragraph 3.1.1: Probably it would be better to overlap the two chromatograms in Figure 1.
*Indeed, we tried to do it, but the larger FMOC peaks make the information not clear. In our opinion, the present picture allows us to make a better comparison among the signals of amines present in the sample.*

Paragraph 3.1.2: Since the derivatization is quite fast already at room temperature and the unreacted Fmoc is a chromatographic interference, why have you used higher concentration of Fmoc compared to previous studies?
*As reported at the end of the section "our sample pre-treatment appears faster as it does not need successive liquid-liquid extraction or preconcentration steps". These significant advantages, with respect to previous published procedures, have been obtained by using an excess of FMOC (which allow us to obtain the quantitative derivatization of all amines, also for DEA) and reducing its chromatographic interferences using acidic eluents and the optimized reaction time/temperature. In our opinion, these concepts are correctly reported and discussed in the text. Small changes make them clearer.*
Using a large excess of Fmoc-Osu, the reaction was quantitative (also for DEA) and very fast even at ambient temperature, as it takes less than 5 minutes to complete.
… carried out at 50 °C, for 10 min, to completely eliminate the chromatographic peak relative to the unreacted Fmoc that coelutes with the derivatized ethylamine.

Paragraph 3.2: Have you investigate the potential effect of the matrix? How other compounds that might be present on the filter could interfere with the extraction, derivatization yield, recovery, etc.?
*The matrix effect was investigated by recovery experiments in which known amounts of amines were added to PM filters. The slope of the recovery functions led to the estimation of quantitative recoveries for the amines of interest. Although it is not a measurement of the analytical bias, this result is a good indication of the negligible effect of the matrix components on the effective recovery of each analyte at various steps of the procedure, as well as on instrument response. This is also confirmed by the equivalent results obtained using an independent (and more selective) detection system (MS). A small change should make the concept clearer.*
… recoveries ranged from 95% to 101%, supporting the hypothesis of a negligible effect (bias) of the matrix components on the effective recovery of each analyte at various steps of the procedure, as well as on the instrument response.

Line 319: It would be interesting to add some info about the type of emission from tanning industry
*Since the 60s, tanneries in the Chiampo Valley (Vicenza, Italy) have been the main responsible for the contamination of both the surface/ground waters and the atmosphere of a larger area. In particular, large*

*amounts of H₂S and VOCs are directly emitted into the atmosphere, with the consequence that a typical odor is always present in the valley. This is confirmed by the Regional Environmental Agency (ARPAV) that recently (2022) measured high atmospheric H2S levels (daily average of up to 241 μg/m³, hourly average up to 1067 μg/m³). Regarding VOCs, emission inventory data (2022) show that in the Montorso municipality (in the middle of the tanning district), VOC emissions from the tanneries amount to 87.5% of total VOC sources. Similar values are estimated for the other municipality of the district. Although the primary aerosol emissions from the tanneries are comparable to those estimated for other sources (traffic, other industries, agriculture, domestic), a significant contribution of the tanning district to the secondary aerosol production in a larger area is predictable. These phenomena are carefully monitored by ARPAV and are detailed in its technical reports.*

*A new sentence in the revised manuscript, with references to the ARPAV reports, summarizes these details.*

In this connection, we underline that since the 60s the tanneries of the district have been the main responsible for the contamination of both the surface/ground waters and the atmosphere of a larger area. In particular, high amounts of H₂S and VOCs (mainly toluene, xylenes, ethylacetate, and butylacetate) are emitted into the atmosphere by this industrial activity. This is confirmed by the Regional Environmental Agency (ARPAV) which recently measured high atmospheric levels of H₂S (up to 241 μg/m³, daily average, and 1067 μg/m³, hourly average) and estimated that 60-88% of the total VOC emitted comes from the tanneries of the district (ARPAV, 2023; JRC, 2013). Primary aerosol emissions from these tanneries are comparable to those estimated for other common sources, but, associated with high H₂S and VOC emissions, a significant contribution to secondary aerosol production in a larger area is predictable. In effect, the significant levels of amines measured here would also be associated with the accumulation of pollutants that affects the entire Po Valley during Winter and Autumn (Masiol et al., 2015).

Paragraph 3.3: I think the comparison between the data obtained in this work and others reported in literature could be also summarized in a table.
*As required, a new table has been added to Supplementary Materials section.*

Technical comments:
*All these comments have been addressed.*

Line 70-71: specify that is not possible separate amines with the same molecular mass without MS2 analysis.
This drawback is also reported in direct HPLC-MS analysis of amines (Kieloaho et al. 2013; Tzitzikalaki et al., 2021), in view of the difficulties in separating these species (non-derivatized, in the absence of a suitable MS2 detection system).

Line 71: DMA/EA PA/TMA acronyms have not been specified before. Please do it.
*Sorry, they are defined in the Abstract*
(i.e. DMA/EA and PA/trimethylamine, TMA)

Line 175: Please use the subscript for the "x" in NOx
NO$_x$

Line 342: please define the acronym OC
between DMA and OC (organic carbon concentration in PM10),

Line 360: Please use the subscript for the "x" in NOx
NO$_x$

Added references

ARPAV: I Monitoraggi della Qualità dell'Aria nell'Area della Concia Anno 2022, 1–54, 2023.

JRC: Best Available Techniques (BAT) Reference Document for the Tanning of Hides and Skins, 290 pp., 2013.

**AR-2023-3, Anonymous Referee #2 Comments and *Reply by the Authors* (*italic*, actions in blue)**
*We thank the reviewer for the detailed comments and suggestions. They have been carefully addressed to improve the manuscript.*

Overall Comments

The Authors present the results of measuring amines by an established liquid chromatography (LC) fluorescence derivatization (FLD) technique that they've optimized for their needs in-house. The performance of the LC-FLD is compared to an LC-mass spectrometry (MS) determination following additions of high quantities of amines to real samples. The Authors conclude that the method is validated, but the analytical approaches are not consistent with validation of a technique. It is likely that they have the required data to properly validate their technique, pending revision, based on the experiments they've stated were conducted in the methodology.

*We underline that the measurements by LC-MS represent a small part of the validation process of the LC-FLD method. Validation procedures and results are reported in Section 3.2; in Section 3.2.1 we compare the results obtained by two different detection techniques, demonstrating that there are not significant differences between them. This result supports the hypothesis that possible chromatographic interferences (bias) of the LC-FLD method have a neglecting effect on the quantitative determination of amines. This important conclusion is clearly stated at the end of Section 3.2.1:*

This result suggests the absence of significant chromatographic interferences (bias) in the optimized UHPLC-FLD analytical method.

The results of the amine determinations are then paired with a number of air quality and meteorological measurements, then apply principal component analysis to perform a basic assessment of associations between these. They survey a small number of other reports from the literature, which could be improved.

*We thank the reviewer for the appropriate criticism. The relevant literature has been cited in the new version of the manuscript, also summarized in Table S6.*

The initial emphasis in the article regarding the potential role of tanneries as sources of amines is not presented in the results and discussion, nor is a compelling set of measurements with respect to this potential amine source shown. It seems that the manuscript, overall, could use quite a bit of attention to detail in the narrative the authors wish to present alongside a more data-driven validation of their optimized analytical techniques.

*We agree, but we would underline that the aim of the work is the development/validation of a simple analytical method rather than a study on sources in the specific area. And in this connection, the application of the analytical procedure in the characterization of the tanning district aerosol was aimed at testing its possible implementation in environmental monitoring campaigns. Some changes in the Introduction and R&D make the manuscript more coherent (see the answers to specific comments below).*

Specific comments provided by line number below should be addressed before this manuscript should be considered acceptable for publication in Aerosol Research.

Specific Comments

Page 2, Lines 43-44: They can also compete with NH3 in these reactions. The work of Bzdek should be mentioned here, as the authors draw on (or sometimes miss) the displacement concept elsewhere. There also needs to be a better job done in describing how gas phase sources of amines translates to their presence in the condensed phase. Reaction schemes that include phase notation could be helpful for this.

*The essential aspects of the formation of ammonium salts have been summarized here. We think that a detailed outline is not consistent with the Introduction chapter. Thanks to the reviewer's comment, replacement processes are now mentioned.*

Theoretical and experimental studies also show that alkylamines can react with ammonium sulfate and ammonium bisulfate clusters resulting in the displacement of ammonium with aminium (Bzdek et al., 2010; Qui et al., 2011). Amines are also believed to play an important role in stabilizing sulfuric acid clusters in the early stages of new particle formation (NPF) (Loukonen et al., 2010; Almeida et al., 2013; Yao et al., 2018; Xiao et al., 2021).

Page 2, Line 45: See work from Denkenberger that performed some of the earliest work on SOA formation from amines.

*OK, a reference has been included.*

Denkenberger et al., 2007

Page 2, Line 48: Toxicity depends on the dose, so this statement should be revised, or moved to a more relevant part of the introduction that discusses point sources where the levels of amines may be toxic.

*We do not agree. Toxicity is an intrinsic property of the chemical substance. Of course, the toxic effect depends on the dose. Please, leave the phrase as is.*

Page 2, Line 52: Just cite the paper. No need for added commentary.

*The disputed part of the phrase has been removed.*

( …)

Page 2, Line 53: 'biomass burning and degradation' are the main continental sources, yes, but the Authors have missed the enormous source of amines in marine environments. Literature survey is not complete and providing correct atmospheric context.

*We thank the reviewer for this comment. Marine emission is now contemplated.*

The main natural sources include biomass burning and biological activity, in particular from marine environments (van Pinxteren et al.; 2019, Xie et al., 2018)

Page 2, Line 56: Should 'consumption' be 'combustion' here?

*Yes, "combustion of tobacco" is more appropriate.*

combustion of tobacco

Page 2, Line 57: 'Tanneries may also represent a significant source'. Be more explicit, since this is a target industry in your work. What particular aspect of the industrial processes here are thought to be (or known to be) using substantial quantities of amines? In the results and discussion, this idea is not revisited, so maybe it is not relevant at all and can be removed from the introduction?

*We complete this sentence as required.*

Tanneries may also represent a significant source (JRC reference reports, 2013) as amines are formed and released in the atmosphere both in the preliminary steps of the tanning process (storage, desalting, and pealing of the row hide) and in the wastewater treatment plants.

Page 3, Lines 67-70: Reading the literature selectively. For example, IC methods have shown substantial improvement. See further work from VandenBoer on this topic, led by Authors Place and Salehpoor, where the noted issues have been overcome. The former work is 6 years old already.

*We were probably too selective; now we have included the relevant references.*

These problems can be addressed by carefully adjusting the temperature and the elution program, or by modifying the retention times of the interfering ions with an eluent additive (Place et al., 2017, Feng et al., 2020).

Page 3, Line 75: This paragraph on atmospheric measurement techniques is missing atmospheric gas sampling of amines by mass spectrometers like the PTR-MS? Not mentioned, but fairly prevalent in the literature. Overall, the other instrumental approaches to measuring amines is not particularly well-synthesize with respect to the advantages and disadvantages each one presents to the atmospheric analytical chemist.

*We agree. PTR-MS is the elective technique for the determination of amines in the gas phase, also at very low concentration levels. Our analysis of the literature was initially limited to the determination of amines in PM, but now, as suggested by the reviewer, we have corrected this lack. As for the comparison of the different analytical approaches, we here present (in Introduction) a brief summary. More data are reported in R&D Sections, specific for the PM analysis.*

PTR-MS is the elective technique for the direct (and on-line) determination of amines in the gas phase (Chang et al., 2022: Wang et al., 2020] with elevated analytical performances (fast response, high sensitivity). Unfortunately, it fails the separation of amines with the same molecular mass (i.e. DMA/EA and PA/TMA). This drawback is also reported in direct HPLC-MS analysis of amines (Kieloaho et al. 2013; Tzitzikalaki et al., 2021), in view of the difficulties in separating these species (non-derivatized, in the absence of a suitable $MS^2$ detection system). Direct analyses of amines can also be performed using GC (Bai et al., 2019). However, the polar nature of small aliphatic amines makes these analyses unfavorable due to the adsorption and decomposition of the solute in the column, resulting in peak tailing and analyte losses.

Page 3, Line 80: 'simple and fast derivatization step'. Be more specific. Are analytes added directly to aqueous extracts and then analyzed? I.e. one pot? How fast? And what is 'mild' in respect to? Temperature, pH, something else? And contrast this against the general properties of the other derivatization methods. Give specific examples.

*Here (Introduction) we summarized the derivatization approach in the analysis of amines, producing the relevant literature. In our opinion, experimental details are not opportune in an Introduction section. Please leave the sentence as is.*

Page 3, Lines 83-84: 'UV-vis absorption and fluorescence'. Need to state specifically why these are advantageous analytical techniques over something like a mass-spec or conductivity detector.

*Some basic aspects of instrumental analytical chemistry should be widely disseminated in our scientific community; we would avoid emphasizing them again. Anyway, a brief clarification has been added.*

Derivatization can also be used to add a chromophore (or a fluorophore) to derivatized analytes, allowing both improved chromatographic performance and the detection by UV-vis absorption or fluorescence (Płotka-Wasylka et al., 2015; Szulejko and Kim, 2014) with significant higher sensitivities with respect to the conventional detection of non-derivatized species.

Page 4, Line 111: BET measures mass loading. Be specific in how this measurement was used alongside your PM10 samples (or if your sample filters were taken directly from the BET sample impaction setup).

*The use of an on-line analyzer is now better clarified.*

PM10 samples were collected for 24 h and directly measured using an automated beta attenuation analyzer (SWAM 5a Monitor, FAI Instruments, Italy) working at a constant flow....

Page 4, Lines 117-118: Are the artifacts in the gas or particle phase? For the Shen reference regarding DEA here: If this is not clear and decisive, then it may be worth doing a more general description of the potential

for analytical bias. If there are different reasons and none are particularly conclusive, then Shen et al could not have made a well described argument.

*In our opinion, it is opportune to mention that the sampling procedure can be affected by artifacts. Shen et al. investigated the relevant mechanisms (for instance, gas/particle phase transfers) to whom refer for details. According to the comment/answer from Reviewer #1, we are confident that in our procedure the artifacts in amines determination were very limited. In fact, we focused on amines determination, but in PM we find their ammonium salts, rapidly formed in the atmosphere by reaction with acids, which are characterized by a significantly lower volatility. To better summarize these aspects, the relevant phrase has been rewritten.*

It is worth noticing that artifacts can occur in the sampling of aliphatic amines, well described by Shen et al. (2017). They are mainly due to gas/particle phase transfers and some structure rearrangements, although different hypotheses regarding DEA are proposed. We did not estimate these artifacts, but considering that amines are present in PM as ammonium salts, the sampling losses of these low-volatile compounds were probably limited.

Page 4, Line 131: '0.1 M HCl' Give rationale. To keep analytes protonated and non-volatile?

*We would avoid "discussion" in M&M section. Moreover, this comment regards the acid-base equilibrium of amines in aqueous solution, a very basilar concept for all AR readers. We also underline that relevant details are reported in Section 3.1.2.*

Page 4, Line 133: The extraction volume of 3 mL is larger than the size of this 1.5 mL test tube? Please revise the procedural details for accuracy.

*We thank the Reviewer for evidencing our trivial error, now corrected.*

1.5 mL of the extract was then filtered (PTFE membrane, 0.2 µm pore size) inside a 1.5 mL test tube …

Page 5, Lines 135-140: Molar concentration is more important to communicate. For example, it would help here to communicate that the derivatization reagent is present in excess. Similarly for the buffer capacity and reaction condition stability molar concentration and pH of the borate buffer should be given. Revise.

*We have now also reported the molar concentration of these solutions, accordingly.*

The derivatization reagent was prepared by dissolving a few mg of Fmoc-OSu in acetonitrile to obtain a concentration of 8.9 mM (3 g/L). Borate buffer (0.13 M) was prepared by dissolving 5 g of sodium tetraborate in 100 mL of milliQ water.

Then 150 µL of borate buffer and 250 µL of derivatization reagent were added to the solution (pH≈9).

Page 5, Line 140: How was the temperature control achieved? Heating mantle? Heating block? Please add.

... using a thermostatic heating block.

Page 5, Line 145: Autosampler details should accompany the injection volume at the start of the sentence.

*We rephrased it, accordingly.*

The solution containing the derivatized analytes was injected into an HPLC (LC-20AD XR Shimadzu; Canby, OR, USA) equipped with an autosampler (SIL-20AC XR Shimadzu, injection volume 5 µL) and a fluorescence detector (RF-20° XS Shimadzu).

Page 5, Line 146: Are absorption/emission profiles for all of the analyte derivatives possible to present in the SI?

*We think this is excessive. All these amine derivatives have similar (identical) fluorescence spectra, due to the amine-fluorophore structure. Reaction scheme, molecular structure of the derivatives, and spectra are well described in the literature.*

Page 5, Line 151: Concentration of B was increased linearly? After a hold at 40% for 1 minute? The gradient program could be written more clearly or placed in a supporting table.

*Yes, linearly. It has been clarified. We would avoid reporting this data in a new table.*

... after 1.5 min the concentration of B was increased linearly to 60% in the span of 13.5 min, ...

... then the concentration of B was raised to 100% in 0,2 min and maintained for 6 min (before returning the system to the initial condition, for a chromatographic run lasting 28 min).

Page 5, Lines 151-152: 'elution mixture' should be 'mobile phase composition'

*The previous answer solves this question.*

Page 5, Line 153: In atmospheric samples, the moles per volume of gas sample are much more useful for the community of researchers. As such, it would be most helpful to have calibration data also presented using units of moles instead of mass, since the ranges are not actually the same for each analyte on this basis. Or at least give the mol/L concentrations for this first standard, so people can get an idea of what this mass per solvent volume value is equivalent to.

*Dealing with PM, mass concentrations are widely used. It is the case of the air quality assessment based (by EU regulation) on concentration limits expressed in mass ($\mu g/m^3$). We suggest to maintain the present units. As suggested, for the concentrated standard solution, molar concentration has been reported.*

Concentrated standard solutions (1 g/L; for DMA it corresponds to 22 mM) were prepared ...

Page 6, Lines 198-199: The reaction scheme/equation for amines, generically, should be presented somewhere, alongside the necessary reaction conditions, as stated above.

*The cited literature contains detailed reaction schemes and conditions here summarized in the text only for the relevant analytical aspects. We think that an extended description of known derivatization reactions is not necessary in the present environmental-analytical contest. A better reference on this topic has been added.*

Details regarding these reactions are reported in the literature (Iqbal et al., 2014; Lozanov et al., 2004; Płotka-Wasylka et al., 2015.)

Page 7, Line 205: 'in about 20 minutes'. Performance is being oversold here. The run is nearly 30 minutes long as per the x-axis in Figure 1.

*Respecting the referee opinion, we would underline that this discussion regards the comparison among different columns tested in the optimization of the method. For all of them, the separation of amines occurred in approximately 20 min, as shown in Figure 1. The chromatographic runs take around 30 min because they include the column cleaning and equilibration steps. All LC users know that these steps are necessary to maintain good and reproducible chromatographic condition. We suggest to maintain this phrase as it is, instrumental analysis time has been specified in the text of the Section 2.4 (M&M).*

... for a chromatographic run lasting 32 min).

Page 7, Line 206: 'of its slightly higher efficiency'. Be quantitative in what you mean by this. Better peak shape? Better N? both? Would be valuable to show the results of the three different separations on a single figure in the SI. A table of performance metrics would also be valuable.

*Taking into account the aim of the Journal, we avoided digressions on marginal aspects. In this discussion, we stressed the fact that other analytical columns can be used with similar chromatographic performances. We do not think that details on theoretical plats or chromatographic resolution could be of utility for future users: in our opinion, the information reported in the text and in Figure 1 represents a good synthesis.*

Page 7, Lines 210-211: Were these by-products identified?? Is the derivatization of NH4+ an asset of this method that should be presented more explicitly? Can these be labeled in Figure 1, alongside the

presentation of the chromatogram for a reagent blank and method blank? This would be useful in communicating how 'clean' the derivatization is versus how important a high-efficiency separation needs to be when implemented alongside this derivatization.

*These species have not been identified here, but their presence is reported in the literature. If some FMOC by-products can be reduced by using our procedure, a large excess of $NH_4^+$ in PM produces an intense peak (overlapped with some by-products peaks) in the first part of the chromatogram. As described in the text, method optimization was also aimed at separating these peaks from the amines signals. As for the $NH_3$ derivative, at the moment we see it as in interfering species (but well solved by LC); a modified version of the present procedure could be of interest for the analysis of samples containing very low amount of ammonia, but probably IC will remain the elective technique.*

Page 7, Line 222: Volume comparison is incorrect to use here. The derivatization depends on the stoichiometry, and here, it seems that you are trying to make sure this is present in greater excess than prior work. That decision needs to be justified using molarity of derivatization agent, and expected range of molarity for analytes in sample extracts at a minimum.

*We agree. The difficulties arise from previous papers that do not report the real concentration of FMOC used in the derivatization step. We have now estimated it and reported in the text, explicitly.*

... of derivatizing agent (250 μL in 700 μL (3.2 mM) instead of just 400 μL in 10 mL (0.36 mM) of Fmoc-OSu in acetonitrile …

Page 7, Line 224: The extraction of amines into water has been long established as a quantitative technique. What was actually done here and where are the analytical results of the different tests for time, temperature, etc using control solutions of known composition? These should be tabulated at a minimum, but could probably use some figures (see further comments below).

*In this phrase/section we mentioned the criteria and tests that we used to reach quantitative recoveries of the analytes. For brevity, only the relevant results are reported here. The validation data for the entire procedure, demonstrating quantitative recoveries, are discussed in the proper section (3.2). In our opinion, this data presentation is more effective than arguing about the characteristics of each individual step.*

Page 7, Line 230: How was this extraction comparison done? Data to prove this? What analytes were assessed?

Page 7, Line 231: What was the criterion to conclude that the reaction was complete? Where is the data of that criterion versus reaction time, and with a trace for each temperature, to clearly depict this?

*Section 3.2 reports all details of the procedure. The present paragraph focused on the variables that have been optimized (extraction time and T) to obtain quantitative results (as reported "to gain the maximum recovery for each of the analytes"). Portion of the same filter (also spiked with all analytes) were extracted under different condition (time) and analyzed. Portions of the same extract (and standard solutions) were analyzed after derivatization at different temperatures. The experimental results were compared with the expected results, concluding what it is summarized in 3 lines. We ask the Editor: is an extended description of these steps necessary and useful for the readers? In our opinion, these lines correctly summarize the relevant information.*

Page 8, Figure 1: Known byproducts should be tagged and described in the manuscript, as well as in this figure. That way, the presence and abundance of otherwise unrecognized amines in the real sample trace can be better communicated through the chemical selectivity of this method. Can panels a and b combined with the traces offset? The axes are also not the same line colour or width, with matched font as the data trace or the labels. Suggest revising for consistency.

*As stated in a previous answer, FMOC by-products have not been identified and were not tagged in the chromatogram. Certainly, the non-amine signals in the standard solution chromatogram (Fig 1a) are not due to unknown amines! So, most of them can be associated to FMOC by-products.*
*Regarding the restyling of the picture, as answered to Reviewer #1, we prefer to maintain the picture in the actual form because "In our opinion, the present picture allows us to make a better comparison among the signals of amines present in the sample". Yes, we will use the same line color for axes and numbers.*

Last, why are the details of the one depicted sample relevant for this plot?
*It is the chromatogram from a real sample that evidences the well detectable amine signals (although PM10 is not excessively high).*

Page 8, Line 239: 'that may represent a serious chromatographic interference'. Another reason to present a trace of FLD Intensity versus retention time for an analytical blank to show this. Can be added to the offset traces in Figure 1.
*Consequently, a new picture was prepared with the chromatogram of a blank sample. We suggest inserting it in the SM, leaving Figure 1 in the present form.*

Page 8, Line 245: 'faster' is not really analytically descriptive. What about fewer sample handling steps, meaning higher method throughput? Derivatization completion is not effectively demonstrated to support this statement, nor is the quantitative capability (i.e. recoveries need to be properly assessed, and/or internal standards used to track derivatization, as per further comments below).
*About the quick derivatization reaction, we have discussed earlir giving details on the quantitative recovery of the both steps (extraction and derivatization). Here, we underline that the method avoids L/L extraction step with obvious advantages with respect other procedures (also in terms of time of analysis). As for bias, all these doubts are clarified in the validation section (3.2). We think it is correct to leave the discussion regarding the analytical bias in the proper section.*

Page 9, Line 248: 'were estimated' Estimation is not appropriate for method validation. They should be (and appear to be). Suggest revising here.
*We agree.*
… and recoveries of the method were measured experimentally.

Page 9, Line 253: 'six amines at 10 mg/L' Is this relevant? The final concentration in the analyzed sample is better to present here. Why couldn't the actual sample and its amine concentrations be used? This would be more reliable in representing reproducibility, as the matrix effects would be properly captured. Understandably, if the entire amine suite is not in the sample, spiking could be a way to evaluate those in the real matrix, but it needs to be clearly presented that the spike levels are reasonably approximating the range expected in real atmospheric aerosol extracts.
*The data reported (15 µL … at 10 mg/L) represents the real treatment we have done. The corresponding concentration in the real sample, already indicated in Table 1, is now reported in the text.*
… each spiked with 15 µL of a mixture containing the six amines at 10 mg/L (corresponding to 4.1 ng/m$^3$ in the sampled PM10 and in line with the concentration ranges measured in the real samples).

Page 9. Line 253: 'the recoveries'. This is a test for matrix effects, not recovery. Recovery refers to spiking a known quantity into a sample and then assessing the amount using your typical calibration approach, then calculating the relative error. Revise.
*This comment is surprising. The reviewer misses that we did exactly what he suggests, but using 3 concentration levels of additions. So, the slope of the recovery functions ($C_{experimental}$ vs $C_{added}$) is the average recovery in that concentration range. Non-quantitative recovery (slope ≠1) would suggest a bias, that might*

be due to a matrix effect and/or to many other causes of systematic error. Fortunately, this is not the case here. For a better understanding, the recovery function has been defined, and a comment has been added.
… determining the slope of each recovery function ($C_{experimental}$ vs $C_{added}$).
… It was found that the recoveries ranged from 95% to 101%, supporting the hypothesis of a negligible effect (bias) of the matrix components on the effective recovery of each analyte at various steps of the procedure, as well as on the instrument response.

Page 9, Line 264: 'also by better accuracy'. Better accuracy has not been demonstrated by any of the presented metrics. Recovery is not an assessment of accuracy. Revise.
*The previous answer could help the present discussion. In absence of a Standard Reference Material we are not able to quantify the analytical bias. But quantitative recoveries (and the comparison of results obtained by an independent detection modality, MS (Section 3.2.1), excluding any chromatographic interference) support the hypothesis that any possible bias, if present, has a neglecting effect on the quantitative measurements. This result, combined with the good repeatability, makes the analytical procedure suitable for the accurate determination of amines in PM. In our opinion, the sentence correctly expresses these important concepts.*

Page 9, Line 266: This discussion is incomplete. Where are the upper and lower range assessments of RSD? These were supposedly done with two different concentration standards, to represent upper and lower limits. How relevant is the calibration range to the quantities found in actual samples? Recovery and matrix effects need to be better explained. Last, the discussion here also needs to indicate the limitations of this approach more fully (complex chromatogram, cannot detect triply substituted analytes, etc.) in the context of the literature, not just other methods reporting use of this derivatization agent.
*We would not enlarge this section that, we underline, delas with Method Validation. In our opinion, relevant results have been reported. Regarding the specific question, we have determined RSD in a real sample (spiked, n=4) at a concentration level in line with the real samples (as correctly reported in the text and in the table). It was not reasonable to divide the sample into more parts to test the precision in a larger concentration range or replicate this procedure using a higher number of real samples (not so easy to collect). As for the other points, in view of the advantage of the proposed derivatization procedure, we mainly focused on the comparison of results with the published methods that used a similar approach and demonstrated similar performances.*

Page 10, Line 277: Was butylamine used as an internal standard in the fluorescence detection as well? Why not use mass-labeled IS here? Is an IS even required in the FLD approach? Could that fact be used as a point of analytical strength versus a much more expensive technique like LC-MS? The use of a common residual fragment here from the derivatization reagent seems risky and the selected qualifier ions for each amine are not presented in a table of the SI to demonstrate that the choice of quantitation ion has been done well.
*We had no advantages in using IS for calibrating the LC-FLD method. The absence of matrix effect or other systematic errors allows us to use an external calibration. As is known, the use of isotopic labelled standards in MS analysis improves the quality of the results, and they surely would have been used in the case of optimization/implementation of a new LC-MS method of analysis. But this more selective technique is adopted here just to investigate on the presence of chromatographic interferences that would introduce bias in the LC-FLD procedure. Moreover, the absence of an amine (BA) in those samples prompted us to use it as IS in the LC-MS analysis. In conclusion, statistically coincident results obtained from independent FLD and MS signals support the hypothesis that unsignificant chromatographic interferences affect the LC-FLD method. In view of the last question, Table 5S in SM has been completed with the m/z values used for the confirmation of the single amines by LC-MS. As reported in the text, the quantification of all amines used the signal at m/z=179.0861.*

Page 10, Line 281: 'measured concentrations of the amine derivatives were statistically equivalent' This does not represent a true orthogonal assessment of method performance, and therefore validation. Spiking the analytes into your samples at high concentrations instead of at their actual environmental levels is a weak evaluation.

*We thank the reviewer for this relevant question. In effect, Table 5S reports the concentrations determined in the solutions extracted from the PM samples (μg/L, clearly indicated in the first column of Table S5). They are comparable to those obtained from real samples. For clarity, both the caption of the Table 5S and the text have been changed.*

**Table 5S** Comparison between the results obtained with FLD and MS analysis of the solutions extracted from three different samples of PM10.

... 10 μL of a mixture containing the amines at 10 mg/L was added to the three samples (corresponding to 1.8 ng/m$^3$).

Page 10, Line 292: This is a self-contradicting statement. Simply say that butylamine 'was not detected'

*A very weak signal of BA is detected both in blank and PM samples, lower than the estimated LOD. The sentence reports this information correctly.*

Pages 10-11, Lines 299-313: Amine mass loadings have been reported in a large number of studies in recent decades. Why are the specific studies cited here relevant for comparison? Suggest revising the writing here to be more specific or more appropriately referencing the literature (i.e. a review paper that broadly discusses atmospheric mass loadings, like Ge et al.). Suggest reducing this discussion of each paper separately since the comparisons that have been selected are arbitrary. It would be more appropriate to conduct a thorough review of comparable and contrastable measurements of atmospheric amines in aerosol samples, supported by a carefully constructed table to present the relevant parameters and amines.

*We respect the opinion of the reviewer, but this is not a review paper reporting and comparing concentrations measured throughout the world. Our discussion has been focused on amines in PM: some relevant data are commented in the text; more details have been presented in a new Table (S6) as required by Reviewer #1.*

These studies are here briefly discussed as examples regarding the alkylamine concentrations commonly found in the PM of urbanized areas; more information is summarized in table S6.

Page 11, Line 319: Agriculture has been well-reported in the Po Valley to be a driver of aerosol composition. Suggest reviewing more of the prior literature from this location, including potential amine sources.

*As evidenced in the discussion with Reviewer #1, a new sentence with references to the ARPAV reports summarizes the available details.*

... such as vehicular exhaust or intense industrial activity (tanning industry, in particular). Another potentially important source could be agricultural activity, which is also strongly associated with ammonia emission (Backes et al., 2016). However, recent studies suggest that in urbanized areas the contribution of non-agricultural sources is dominant (Chang et al., 2022). In this connection, we underline that since the 60s the tanneries of the district have been the main responsible for the contamination of both the surface/ground waters and the atmosphere of a larger area. In particular, high amounts of H$_2$S and VOCs (mainly toluene, xylenes, ethylacetate, and butylacetate) are emitted into the atmosphere by this industrial activity. This is confirmed by the Regional Environmental Agency (ARPAV) which recently measured high atmospheric levels of H$_2$S (up to 241 μg/m$^3$, daily average, and 1067 μg/m$^3$, hourly average) and estimated that 60-88% of the total VOC emitted comes from the tanneries of the district (ARPAV, 2023; JRC, 2013). Primary aerosol emissions from these tanneries are comparable to those estimated for other common sources, but, associated with high H$_2$S and VOC emissions, a significant contribution to secondary aerosol production in a larger area is predictable. In effect, the significant levels of amines measured here would also be associated with the accumulation of pollutants that affects the entire Po Valley during Winter and Autumn (Masiol et al., 2015).

Page 12, Lines 335-338: This argument is incorrect. Sulphate must first be neutralized by reduced nitrogen before nitrates can be incorporated significantly into secondary aerosol. This has been established in textbooks for a very long time. Since literature is not being cited, the Reviewer is speculating that the Authors are referring to some recent high impact reports about new particle formation coupling reduced nitrogen species and nitric acid, which was not shown to partition a substantial mass to the condensed phase.

*We agree, it badly reflects the lower concentration of sulfate than nitrate we found in PM. The phrase has been changed.*

… as a result of acid-base processes that lead to the formation of sulfates and nitrates.

Page 12, Line 344: Need to reference the literature here.

*Pertinent literature is here cited.*

(Scotto et al., 2021; Masiol et al., 2020)

Page 12, Lines 345-353: This paragraph starts off alright, but then becomes disorganized and difficult to follow, with lots of speculative statements. Suggest revising to make this more concise and contextually broad to lead into the use of PCA for other amines other than DMA (e.g. cut the statement about calcium, also the discussion of the PCA results, since that is the subject of the next section). Also, take some space to communicate the strengths and limitations of source apportionment techniques, so that your results can be contextualized.

*A more concise and clearer sentence is proposed.*

 These strong correlations have not been found for some amines. For example, significantly lower correlations of EA with both DMA and PM10 were estimated, possibly due to a specific source of EA (for example, traffic combustions) that differs from those that characterize other amines (biomass combustion, tanneries). Of course, this aspect is of great interest and needs further investigation. In this connection, the acquisition of more complete datasets (Masiol et al., 2020) that could include both traffic biomass combustion markers and the use of adequate source apportionment methodologies (PMF) will provide more detailed results regarding the amine sources of this area.

Page 12, Lines 354-364: Missing here is that the meteorology (T, RH) and concentration of the direct precursors (e.g. NH3 and HNO3) dictates their partitioning to the condensed phase on timescales much shorter than those at which the observations were made (i.e. minutes to hours, versus days). This discussion is rather superficial and does not do a good job relating the PCA results to known phase-transfer mechanisms that underpin aerosol production and growth.

*This is a study on an analytical procedure with a preliminary application to a local site; it is not a deep study on phase-transfer mechanisms. We agree that gas species show a different behavior than those present in the particles; in fact, the PCA was performed using PM composition data and taking the former (and meteorological variables) only as auxiliary variables. In our opinion, this brief sentence correctly summarizes the PCA results, confirming differences between two groups of amines detected in the tanning district.*

Page 13, Line 371: Conclusions and abstract should be entirely revised based on the revised contents of the manuscript.

*We think that the comments/questions posed by the Reviewers have been carefully addressed and only minimal changes are necessary to the Conclusions/Abstract sections.*

… in the tannery district (Chiampo Valley, Italy) are comparable to those quantified in other urbanized sites, possibly associated with the presence of important local emission sources …

Minor changes in Supplementary Information/Materials

Table S2: Need to indicate in the caption how measurements below the LOD were treated and presented here.
*Many thanks, we found trivial errors in this table, now corrected.*
**Table S2** Concentrations (determined by IC) of the principal cations and anions contained in PM10 samples.

Figure S2: Why is no time series of the amine measurements for the three different sites being shown alongside other important components that dictate their presence in particles?
*Because an illegible large table would result. We added the site specification in all these tables.*

Table S5: These are spiked values. The sample concentrations should have been determined by using the standard addition approach, not by comparing the concentrations obtained after adding an easily detected concentration to the samples. This is not realistic.
*This comment has been already addressed; please, see the answer to the main comment of the Reviewer.*

Table S6: Table formatting is inappropriate for a manuscript, even in the SI. Revise for proper high quality presentation of results.
*Ok, we modify it.*

---

## Author Response (AR2)

**AR-2023-3, Topic Editor's Comments and *Reply by the Authors* (*italic*, actions in blue)**
*We thank the Editor for the comments and suggestions aimed to improve the manuscript. They have been carefully addressed, accordingly.*

Sampling artifacts
Responding to comments by both reviewers about potential sampling artifacts, you state that "amines are present in PM as ammonium salts". What is the evidence for this statement? What about aminium salts? Regarding the sampling artifacts, ammonium salts such as ammonium nitrate would be prone to volatilization losses during sampling, and it is not convincing to expect limited sampling losses of semi-volatile compounds. Moreover, the sampling artifacts discussed in Shen et al. (2017) are positive artifacts due to adsorption from the gas-phase or salt formation on the filter, which is not discussed at all, and particle bounce-off in impactor sampling (not relevant for filter sampling).
*We apologize for the incorrect name adopted for these chemicals. Actually, they are aminium ions, now correctly reported. Regarding the sampling artifacts, they are reported only by Shen et al. (2017); Szulejko et al. (2014) only mention artifacts in the SPME step of the analytical procedures. Moreover, in the recent work of Chen et al. (2022), the determination of amines in the gas phase appears to be not interfered by the PM sampled (at the same time) on quartz filters. In our humble opinion, the procedure adopted by Shen for the estimation of these artifacts (on filter sampling) is open to criticism (for example, only positive biases were assessed, avoiding investigation on the negative ones; formation of aminium salts on the quartz filter during the sampling is described by the same mechanisms that occur in the atmosphere). Since these aspects have not been explored in depth (neither by Shen nor by us), we suggest caution in the relevant statements, inviting more in-depth analytical studies on the matter.*
It is worth noticing that artifacts can occur in the sampling of aliphatic amines, as described by Shen et al. (2017). They are mainly due to gas/particle phase transfers and some structure rearrangements, although different hypotheses regarding DEA are proposed. However, in studies in which gaseous amines are sampled on acidified quartz filters, significant absorption on untreated filters used for the PM sampling seems to be excluded (Chen et al.; 2022). We did not estimate these artifacts, but considering that amines are present in PM as aminium salts, the sampling losses of these low-volatile compounds were probably limited. In this regard, more in-depth analytical studies would be desirable.

Extraction of amines
With respect to a comment by reviewer #2 about the extraction of amines, please add exemplary figures to the supplementary material showing the instrumental signals for 20 min and 10 min extraction times, and for the four different temperatures tested. These additional figures in the supplementary material will be very useful to the reader.
*Signals from different real samples extracts are not directly confrontable because they refer to different amine concentrations. Regarding the extraction time, the new Table S5 shows the signals obtained under different conditions. As for the derivatization temperature/time, the results of a significant test are now reported in Table S6. The text of the manuscript has been modified, accordingly.*
… reducing it to 10 min (Table S5). The derivatization step was tested at different temperatures (25, 50, 60 °C) and reaction times (5-20 min) (see Table S6 as an example).

Additional comments
In the abstract, with respect to comments by reviewer #2, please change lines 22/23 to "The procedure has been optimized obtaining very satisfactory analytical performances...".
*Done*
… optimized  obtaining …

In line 55, please remove "well recognized in the review of" from the text.
*Sorry, now it has been removed.*

In line 125, please clarify by changing to: "PM10 samples were collected for 24 h, and PM10 mass concentrations were directly measured..."
*We agree, text has been modified, accordingly.*

In line 171, the duration of a chromatograpic run is given as 28 min (a chromatographic run lasting 28 min), while in lines 230/231 the duration is given as 32 min (a chromatographic run lasting 32 min). Please reconcile.
*Sorry, the chromatographic run lasts 32 min.*

In lines 227/228, with respect to comments by reviewer #2, please remove the following part of the statement: ", being able to separate the analytes in about twenty minutes using an acidic mixture of water/acetonitrile as the eluent". In line 229, the qualification "slightly higher efficiency" is unclear to me. Please clarify the advantage of the selected column, or alternatively remove the comparison of the three columns.
*According to the Editor's comment, the phrase has been modified but maintaining information on the similar performances of the tested columns.*
Three RP-C18 columns were preliminary tested for the UHPLC separation of the six derivatized aliphatic amines (Shimadzu XR-ODS III C18, Phenomenex Luna Omega polar C18, Phenomenex Kinetex C18) showing similar performances in terms of selectivity. The Luna Omega column was chosen because of its slightly higher efficiency (better peaks shape).

In line 252, with respect to a comment by reviewer #2, please add possible reasons why the addition of 0.1 M hydrochloric acid might have shown better recoveries, e.g. higher fraction of protonated analytes.
*Acidic conditions are adopted in all extraction procedure described in the literature. Even if the presence of stable metal-amino complexes in PM might be hypothesized (if of interest, see our previous works in this field) for which the acidic water solution promotes their hydrolysis and the solubility of the ligands, it is probable that this effect is simply due to the higher water solubility of the aminium ions with respect to the corresponding amines.*
… in terms of recoveries (from 80-95% to >95%), due to the higher water solubility of the aminium ions with respect to the non-protonated amines.

In line 273, with respect to a comment by reviewer #2, please remove the following part of the statement: "appears faster as it"
*It has been removed.*

In line 298, with respect to a comment by reviewer #2, please remove the following part of the statement: "better accuracy, with"
*We accept this compromise; it has been removed.*

In lines 327/328, with respect to a comment by reviewer #2, please change to "while butylamine was not detected" or alternatively "while the butylamine signal was below the limit of detection".
*We agree, the phrase has been modified, accordingly.*
… while the butylamine signal was below the limit of detection and …

In line 360, please change to "since the 1960s".
*Changed*

With respect to the comment by reviewer #2 about Table S2 in the original supplementary material, how were measurement values below the limit of detection treated and presented in Table S2?
*Table S2 contains some data below LOD (only for sulfate, and all close to LOD). For these concentrations, the values were taken as they are (without empirical correction) in the statistical analysis. This information is now reported in Table S2.*

Concentrations below LOD (sulfate) were taken as they are (without empirical correction) in the statistical analysis.

With respect to the comment by reviewer #2 about time series of the amine measurements for the three different sites: In my opinion it would be valuable to show a time series plot with time on the x-axis and the concentration of the five amines on the y-axis as an addition figure in the supplementary material.
*The required figure (S3) has been added to the supplementary material.*

---

## Author Response (AR3)

**AR-2023-3, Topic Editor's Comments and *Reply by the Authors* (*italic*, actions in blue)**

Thank you for your revised manuscript where you address the open issues raised by the reviewers and myself. In light of the referee reports and your revisions, I am delighted to say that your manuscript can be published after one minor technical correction. In line 105, "ammonium salts" should be replaced by "aminium salts".

*Once again, we thank his editor for the pertinent comment. The manuscript has been corrected accordingly.* (as aminium salts).